# An Integrative Review Considering the Impact of Storytelling and Sharing Interventions in Stroke

**DOI:** 10.3390/bs11060088

**Published:** 2021-06-16

**Authors:** Rana Alawafi, Sheeba Rosewilliam, Andrew Soundy

**Affiliations:** 1School of Sport, Exercise and Rehabilitation Sciences, University of Birmingham, Birmingham B15 2TT, UK; RXA904@student.bham.ac.uk (R.A.); S.B.Rosewilliam@bham.ac.uk (S.R.); 2College of Health and Rehabilitation Sciences, Princess Nourah Bint Abdulrahman University, King Khalid International Airport, Riyadh 13412, Saudi Arabia

**Keywords:** stroke, storytelling, narrative, peer support, group dynamic, group process, group sharing, psychosocial well-being, emotion, integrative review

## Abstract

Background: Review-based research is needed which can establish the psychosocial outcomes and mechanisms of “storytelling and sharing” interventions for people with stroke. This information will act to inform the value and development of such interventions. Methods: An integrative review was conducted in three stages: (a) a systematic search strategy was undertaken to focus on articles between 2009 until January 2020 to locate articles the considered storytelling and sharing interventions for people diagnosed with stroke; (b) critical appraisal was undertaken to assess study quality; and (c) synthesis within three stages including data reduction, data display and conclusion. Results: Fourteen articles (including 727 participants) were identified that met the eligibility criteria. Five themes were identified that represented the outcome and mechanisms that appeared to be associated with a stroke intervention. These included introducing the concept of hope and learning to be positive, the enhanced ability to cope, the impact of loneliness and social interaction, impact on emotions, depression and related emotions such as fear. Conclusions: Storytelling interventions appear to impact loneliness, introduce positivity and hope and enable coping through knowledge exchange. The main mechanisms which appeared to influence these outcomes were social comparisons and social control.

## 1. Introduction

The experience of a stroke has a profound effect on negative mood states, social identity and relationships [1]. For instance, within the first two years following a stroke, around a third of people with stroke report depression and around one fifth report anxiety [2]. There is a clear need for psychosocial interventions which provide emotional, informational and motivational social support [1]. Good psychosocial rehabilitation is associated with better health outcomes and community reintegration for people with stroke [3]. Storytelling is a recent behavioural-based intervention that can address the psychosocial needs of people with stroke. The process of storytelling and peer sharing (a term used to represent sharing of information with a peer which often involves personal stories; where stories is the most common ‘vehicle’ used to share experiences) can have a multitude of beneficial effects identified across people with stroke and brain injury [4,5]: (1) being able to identify personal experience that others hear freely; (2) being able to release emotions through stories; and (3) being able to learn and reflect on your own story. As an essential part of life, storytelling is a natural way of expressing who we are and what values we hold. It is one of the most natural ways of giving meaning to the experiences of illness [6]. Traditionally, storytelling has been undertaken for multiple purposes including helping others understand dangers, transferring knowledge, entertaining and maintaining cultural heritage [7].

Literature reviews that consider storytelling interventions across chronic illness groups have found consistent evidence which illustrates a number of benefits including; an ability to create a collective group voice and learning, ability to mobilise the group’s mental well-being, reduced levels of depression, increased socialisation, and increased levels of perceived quality of life [8,9,10]. To the best of the authors’ knowledge, no integrative review has been conducted to consider the value and impact of ‘storytelling and sharing’ on individuals with stroke. The principle aim of this integrated review is to identify evidence of the impact of storytelling/narrative or sharing interventions on managing the psychosocial challenges after stroke and to document the outcomes. In addition, it aims to explore the experiences of stroke survivors and other stakeholders (carers, family, health care professionals or other staff) participating in “storytelling and sharing” in order to identify the mechanisms underpinning specific outcomes.

## 2. Materials and Methods

An integrative review framework [11] was selected. Three stages were performed: (1) eligibility criteria and literature search process; (2) data evaluation; and (3) data synthesis. The presentation of the review was supported by the ENTREQ [12] (see Appendix A). A PRISMA flow diagram was used to document the output of search results [13]. A subtle-realist paradigmatic position was adopted. This recognises the individual nature of experiences while seeking to identify “common realities” that individuals can relate to. The common features across the included studies are the main focus of this review.

### 2.1. Protocol and Registration

A protocol was registered with PROSPERO with the following ID: CRD42020160984.

### 2.2. Eligibility Criteria

The PICOS (participants, intervention, comparison, outcomes and study design) standardised framework was used to determine eligibility for inclusion in the review.

#### 2.2.1. Participants

Studies were eligible if they had used people with a clinical diagnosis of stroke. The standard WHO definition of stroke was adopted: a “*clinical condition characterised by sudden and rapidly developing signs of focal or global interruption of cerebral functions that should last more than twenty-four hours or could lead to death, with no evident cause other than vascular origins*” [14]. Participants who had been diagnosed with any of the three main subtypes of stroke, namely ischemic stroke, intracerebral haemorrhage and subarachnoid haemorrhage, were included. No limit was identified for the time post stroke, because of the lack of evidence around using storytelling and because past reviews [8,9,10] have identified consistent results which are likely common across participants. Participants who had been diagnosed with transient ischemic stroke (TIA) were excluded, as TIA lasts less than 24 h and often does not lead to any long-term impairment [15]. Participants must have been aged eighteen or older and from any gender or ethnic group. The opinions of multiple cohorts of the intervention of interest, for example healthcare professionals, carers and family members, were included. Where varied population groups were included, separate analyses of findings for participants with stroke were needed.

#### 2.2.2. Intervention

The core intervention under assessment had to contain storytelling or narrative-based approaches that involved the following: sharing stories or peer sharing or interactive discussions occurring in a group that involved exchanging illness-related stories or experiences or challenges or sharing information related to diagnosis of and living with stroke. The main component of the intervention of interest was contact between at least two stroke survivors in order to capture the psychosocial content of the experiences/narratives. Different formats of intervention were accepted for instance including; Poetry, body mapping, visual arts, and theatre. Different forms of delivery were acceptable including digital (online based) or in person. No specification was made as to the delivery format, e.g., one-to-one activities, group-based work or telephone/internet delivery. The aim of the study had to involve examining and documenting the impact or value of these interventions in assisting stroke survivors to cope with psychosocial and emotional challenges after stroke. Articles that examined the individual experiences of stroke patients participating in interventions where sharing or discussion occurred were also included. Interventions led by peers, by healthcare professionals and by any combination of varied stakeholders were accepted. Articles that explored only education-based interventions or articles that did not employ a particular intervention, for instance narrative analysis, or explore the experience of stroke were excluded.

#### 2.2.3. Comparison

Any active control or inactive control groups were accepted, while studies that did not use any control groups were also included in this review.

#### 2.2.4. Outcomes

Studies using any type of data collection method were included; however, it had to record the effect or perceived value and experience of engaging in the intervention in assisting stroke survivors or carers or families to cope with psychosocial or emotional challenges post stroke. The primary outcomes of qualitative studies were perceived benefits or experiences or phenomena in relation to participating in the intervention of interest. The primary outcomes of the quantitative studies were measures that captured the impact of the intervention on psychosocial and emotional well-being.

#### 2.2.5. Study Design

Any study design was acceptable. Both qualitative, quantitative and mixed method studies were included. Qualitative studies included, but were not limited to, observational cohort, phenomenology and narrative studies. Quantitative studies included but were not limited to randomised controlled trials (RCTs), cluster RCTs, pilot RCTs, quasi-experimental and pre/post-test studies. No restriction was made on study language, studies were translated using Google translate and assessed for clarity by the research team before inclusion. Theses, conference abstracts and review studies were excluded.

### 2.3. Literature Search

A comprehensive and precise electronic literature search was conducted in line with standardised guidelines in order to increase the rigour of this review and minimise incomplete and biased searching [11,16].

The primary researcher and corresponding author conducted electronic searches using a combination of the identified keywords and subject heading/MeSH across the following subject-specific electronic databases: the Cochrane stroke group’s specialised register of trials, MEDLINE, CINAHL, AMED, EMBASE, ZETOC, the ProQuest Nursing and Allied Health Database, PsycINFO and SCOPUS from inception of database until 16 January 2020.

A variety of key words were used: Stroke* OR Cerebrovascular Disease* OR CVA AND Storytelling OR Storytelling intervention OR Story* OR Narrative OR Narration OR Narrative Intervention OR Narrative Interview OR Illness -narrative OR Story-Sharing OR Group-experience OR Group circle OR Talking circle OR Peer group OR Peer support OR peer OR Group education OR Expression OR Emotions OR Self-management OR Self-care OR Health education OR Patient-self management.

The primary researcher then conducted supplementary searches. This included; (a) the first 20 pages of Google Scholar and Science Direct. Additionally, (b) the reference lists of retrieved studies or reviews.

### 2.4. Study Selection

In the initial stage of the selection process, the primary researcher and corresponding author scanned the titles and abstracts of all of the citations for eligibility, removed duplicate studies and identified studies that potentially met the eligibility criteria. Once all potential papers had been identified and selected, the researcher retrieved and read the full text to determine which of them truly met the inclusion criteria. A third author (SR) was identified to resolve any disagreements for inclusion.

### 2.5. Data Extraction

The primary researcher used a predefined data extraction form for each individual study included in order to extract essential study-design related data and demographic information as follows: (1) participants’ demographical variables (age, gender, stroke type and location, time since stroke, co-morbidity/severity of stroke, marital status, living conditions and geographical location), and (2) design related information (methodology aim, eligibility criteria, data collection methods and analysis).

### 2.6. Data Evaluation

The primary author undertook a quality assessment of all articles. This included using a qualitative assessment tool and quantitative assessment tool. The quality of each qualitative study was critically assessed using the 13-item COREQ framework [17], adapted from the 32-item consolidated criteria for reporting qualitative research (COREQ) [18]. This assessment demonstrated the limitations in reporting [19]. The scoring of the studies was used for the certainty assessment. One study was examined in this way yet still included because of the insight given in the results section. This was used as a simple way to include or exclude articles [19]. The Cochrane risk of bias tool was applied to assess the internal validity of each quantitative and mixed-method study [20]. The results were presented to the corresponding author to check. Disagreement were resolved by the third author.

### 2.7. Certainty Assessment

Both the primary and corresponding author undertook the certainty assessment. The strength of evidence and quality scores for the quantitative research was linked to classes of evidence [21]. The confidence of evidence and quality scores for the qualitative research was assessed using the CerQual assessment tool [22]. These assessments were mapped on the results themes to provide an indication of confidence in findings. See Appendix A for details.

### 2.8. Data Synthesis

Data analysis was guided by the general principles of integrative analysis, which require that data are ordered, coded, categorised and summarised. The overarching stages specified by Whittemore and Knafl [11], namely data reduction, data display and conclusion drawing, were followed. (a) The data reduction stage requires a classification of data into types. This was achieved by using a qualitative-led open coding approach [23], which involved immersion in data, and tabulation of results [24]. Next, mind-mapping [25] of the results was undertaken; this involved juxtaposing the results [26] and determining how studies were related to one another. (b) Data display and comparison involved taking data from individual sources and displaying them in terms of variables and subgroups, then considering the existence of patterns, relationships or themes. In order to achieve this, a thematic map based on the mind map was produced and results of studies were added to it in an attempt to saturate each theme [27]. At least two studies were needed to support any subtheme or code in order for them to be included at this stage. This achieved data reduction, as required by Whittemore and Knafl [11] during this process. (c) The final stage, of conclusion drawing, verification and presentation of the integrated synthesis, determined the details of data display. This was achieved by integrating findings together into a qualitative synthesis of findings. Reduction and focus were given towards psychosocial outcomes and mechanisms that were linked to health [28]. Appendix A and onwards in the Appendix A provides full descriptions of data synthesis process.

## 3. Results

### 3.1. Search Output and Studies Included

The initial literature search of electronic databases resulted in 756 potentially relevant studies. Following screening 190 were selected to consider. Fourteen studies were included in the integrated synthesis. This included six qualitative studies [29,30,31,32,33,34], two RCTs [35,36] and six mixed-method studies [37,38,39,40,41,42]. Figure 1 gives a full breakdown of the process of searching and identifying studies.

### 3.2. Study Demographics and Characteristics

#### Participants

Demographic information and participants’ characteristics per study are presented in Appendix A. A total of 727 participants (657 stroke survivors) participated in the studies. The smallest study had a sample size of nine [31] and the largest had 216 [36]. Subjects ranged in age from 20 to 95 years. More than half of the studies recruited both males and females [32,33,34,35,36,37,39]. Three of these [33,35,39] had significantly more males than females.

Time since stroke onset ranged from two weeks post stroke to 174 months post stroke. Participants in the studies with the largest populations [32,35,36,38,40] had ischemic stroke. The type of stroke in the remaining studies [29,30,31,33,34,37,39] was unknown.

Seven studies included stroke survivors who suffered from one or more of the following impairments: mobility problems, motor impairments, balance problems, cognitive problems, fatigue, communication and swallowing difficulties and mood disorders [29,31,33,34,35,41,42]. Kessler et al. [32] included stroke survivors who had moderate dependence when performing basic activities of daily living. Appalasamy et al. [36,40] included participants with several stroke risk factors and most had primary hypertension. The remaining studies did not provide information regarding their participants’ health or characteristics [30,37,38,39].

Half of the studies used subjects who lived in their own homes and the majority lived with a spouse or other family members [29,32,33,35,38], while the remaining studies did not disclose details of participants’ living environment [30,31,34,35,37,39,40,41,42]. Among the 727 participants, 164 were married, 51 had significant others, seven were single, five divorced and one widowed, while the marital status of the other 499 was not disclosed. The details of the intervention (including duration, frequency, format of session) are included in the Appendix A.

### 3.3. Critical Appraisal of Research

The main researcher independently critically assessed the quality of the included studies. No study was excluded due to quality. The Appendix A gives full consideration and a breakdown of data assessment by study type; see Appendix A for qualitative appraisal and Appendix A for quantitative appraisal.

#### 3.3.1. Qualitative Study Appraisal

The lowest scoring study was Hancock [31] that scored 2/13. This study was examined for fatal flaws but specific strengths were identified including; (a) well developed interview schedule and piloting (b) critical insight in findings, verbatim quotes, and (c) findings which were consistent with other studies. Overall weaknesses in reviewed qualitative studies included: (1) lack of reflexivity, making it difficult to assess how the researchers’ personal characteristics and their relationships with participants might have impacted the process and credibility of the results [30,31,32,34] and (2) no consideration of sample size [30,31,33,34]. These findings were used to inform the methodological limitations identified in CerQual [22].

#### 3.3.2. Quantitative Study Appraisal and Risk of Bias

The Cochrane risk of bias tool was used to assess quantitative (n = 3) and mixed methodology studies (n = 5) [20]. The following high risks of bias were identified: (1) no protocol (n = 6), (2) no randomisation procedures (n = 5), (3) no allocation concealment (n = 6). Four of the eight studies had five domains identified as at high risk of bias [37,39,41,42]. Assessment of the components of risk of bias are outlined and described in Appendix A.

#### 3.3.3. Classes of Evidence for Quantitative Research

In accordance with the definition of classes of evidence [21], three studies [35,36,40] were assessed as moderate at low risk, rated II, one [38] as at moderate high risk, rated III, and four [37,39,41,42] were evaluated as at high risk of bias, rated IV.

### 3.4. Synthesis of Research

This subsection reports five themes which provide a narrative synthesis of evidence with considerations given to reported outcomes, summary of evidence and possible or likely psychosocial mechanisms that could explain outcomes. A summary of evidence is given in Table 1.

#### 3.4.1. Theme 1: Introducing the Concept of Hope and Learning to Be Positive Regarding the Future after Stroke

##### Reported Outcomes

The intervention, in both individualised and group formats, appeared to help in introducing the notion of hope after the diagnosis with stroke and helped stroke survivors to be positive regarding the future [30,31,32,34,40]. In the study by Morris & Morris [34], five stroke survivors and their partners of care (n = 5/10) and six peer supports (n = 6/8) agreed with the statement “*Things seem more hopeful since joining the group*” and all participants apart from one peer supporter (n = 17/18) agreed that “*the group helps me feel more positive about my future*”. This was quantified by Chow [35], who reported that stroke survivors in the narrative therapy group demonstrated a significant improvement in the Herth Hope Index score over the control group who received psychoeducational support and the improvement continued four months after the intervention (*p* ≤ 0.05). All participants apart from one stroke survivor and one partner of care (n = 16/18) agreed that “*the group inspires me about the future*” [34].

##### Mechanism Suggested by Evidence

Social comparison appeared to be the primary mechanism through which hope and a perception of positivity increased within interventions. This was achieved in 3 ways; (1) Speaking with an individual who had gone through a similar situation and taking time to listen and share stories was identified as validating the feeling of suffering [32]. Listening to the recovery stories of other survivors could led to the realisation that “*there is life after a stroke*” and that recovery is possible, which was unknown before [31]. (2) The ability to share their own story was particularly valuable to participants [37]. Being in a group with other stroke survivors who were in a similar situation and observing how they improved created a sense of hope. The group context created a space for individuals to share how they successfully managed difficult situations after their stroke, which helped others to learn how to manage their own situations [31,33,34]. (3) Participants identified the ability to transfer the experiences shared into management of their own lives [34].

#### 3.4.2. Theme 2: The Enhanced Ability to Cope

##### Reported Outcomes

Muller et al. [39] reported that the majority of participants (n = 10/13) agreed or strongly agreed with the statement “*I learned new ways of coping with issues related to stroke*”, while more than half (n = 8/13) stated that they used the strategies they had learned outside the group context. Participants in the study by Kirkevold et al. [33] reported that both the individual and group-based dialogue interventions supported their efforts to cope and that this help was important, particularly in the first six months. The dialogue helped participants to describe the challenges such as daily activities and emotional and social situations with which they had to cope, highlighted their coping options and supported them as they tried different strategies and as they examined unanticipated situations [33,34]. Appalasamy et al. [40] identified that narratives promoted a proactive stance and action towards the preventative treatment about stroke. Using humour and being positive in the groups were also commonly identified as a significant component of successful recovery [31].

In a post-participation survey, Muller et al. [39] tracked engagement in activities and exercises that were introduced in the group modules. They found that more than half of participants (n = 8/13) began to participate in different leisure and daily activities beyond the group setting. However, other evaluations of the intervention showed that only the handicap domain of Stroke Impact Scale (SIS) and the home integration domain of the CIQ reached significance: *p* = 0.034 and *p* = 0.002 respectively. Both domains focused on performing basic activities of daily living within the home environment, whereas the SIS self-perceived recovery domain and the social and productivity integration domains of the CIQ did not reach significance. Corsten et al. [41] identified a significant difference in quality of life reporting (a) benefits to physical (*p* = 0.037) and psychosocial (*p* = 0.00001) complaints and (b) reduced perceived physical (*p* = 0.009) and psychosocial (*p* = 0.001) burden.

##### Proposed Mechanism

Taking time to listen and share stories with peers who had positive experiences of recovery helped survivors to understand how others had managed and to identify what was possible through rehabilitation in a positive way, which strengthened motivation towards recovery and aided psychological adaptation [32]. Gaining stroke-related knowledge not only enabled participants to be informed and helped them to understand their diagnosis, but also equipped them with “concrete skills” that they then transferred outside the group context [30,31]. Sources of self-efficacy may be important to this. One study [36] identified a significant (F = 12.41, *p* < 0.001) increase in self efficacy when compared to a control group. Alternatively, humour may allow participants to view stroke in a different way when the experience has been experienced as a burden.

#### 3.4.3. Theme 3: Impact on Loneliness and Social Interaction

##### Reported Outcomes

Engaging in the interventions of interest seemed to aid social well-being. The interventions counteracted a perceived sense of isolation and loneliness [29,30,31,32,33,34,38]. Having time to share, listen and relate to others who had been through similar experiences decreased the feeling of loneliness [29,30,33,34]. In the study by Hancock [31], nearly all nine participants stressed the significance of being with other stroke survivors. Additionally, at least half of participants appreciated the hospital visits or phone calls they received from peers [31]. Participants felt alone and the interventions were something to occupy their time after the incidence of stroke [33,38].

In the process of participating in groups, individuals gained new friendships [30,31,34]. The group provided an opportunity to make new friends in common, to whom participants could relate [34]. One participant reported that other stroke survivors in the group had become friends because they took the time to talk and listen to her, in contrast to other people who might say “*I can’t be bothered*” [31]. One study quantified this by noting that a majority of participants (n = 10/13) agreed or strongly agreed that “*I have made new friends*” [39]. Although, participants in the Masterson-Algar et al. [38] identified a danger of friendships that were built up and then suddenly lost post-intervention. This was quantified by Muller et al. [39], who found that more than half of participants (n = 8/13) reported socialisation as the most important part of attending the group. Nearly half of the participants (n = 6/13) reported having communicated with other participants beyond the group setting by various means including text messages (n = 2/13), Facebook (n = 2/13), email (n = 5/13), telephone (n = 6/13) and actual or planned personal meetings (n = 8/13). However, statistical evaluation of the intervention showed that the social integration domain of the Community Integration Questionnaire (CIQ) did not reach significance (*p* = 0.148).

##### Mechanisms Behind Evidence

Exchanging experiences with other stroke survivors who had gone through similar situations was identified by one participant as inspirational and essential, in particular when facing something as serious as the incidence of stroke [33]. Peers were also seen as a source of inspiration to seek a peer role in the future, which facilitated social identity development [32]. A majority of facilitators (n = 6/7; including health care professionals and family) reported that sharing stories with the group helped them to understand participants “*as people*” [37]. Groups became long-term social networks and support resources for participants and continued validating the sense of suffering outside the group context [30]. Getting to know new peers was likely associated with less distress of being alone. The groups also provided an opportunity to revisit and revise their own story. The unique group environment allowed individuals to present their own story, reflect on it and listen to stories from similar others which provides an opportunity to modify their story.

#### 3.4.4. Theme 4: Impact on Emotions

##### Reported Outcome Depression and Related Emotions

Chow [35] measured depression, and found baseline to one month was insignificant increase in depression (estimated effect size 0.05). However, a significant reduction at 2 months (estimated effect size 2.24) and four months (estimated effect size 2.04). Gurr [37], identified an insignificant decrease in the mean depression score on the Hospital Anxiety and Depression Scale (HADS) after participating in the intervention group (mean pre-group score approximately 8.5; mean post-group score just above 7.5). Corsten et al. [41,42] across both studies identified a positive and significant change for feeling less tired following the group (*p* = 0.06; *p* = 0.00001), this equated to a very large effect size (*d* = 1.02) [41] and small effect size (*d* = 0.30) [42]. Despite further insignificant results from Corsten et al. [41], moderate to very large effect sizes were noted across the following moods; confused (*d* = 1.26), sad (*d* = 0.89), angry (*d* = 1.07), and tense (*d* = 0.68).

Qualitative data identified that sharing experiences with other stroke survivors and validating one’s own personal experiences played a role in reducing the depression commonly experienced by participants [30,31]. Moreover, two participants identified being occupied by attending the monthly scheduled peer groups and meeting stroke survivors who had disabilities, yet displayed positive attitudes, as factors which helped them in overcoming their depression after stroke [31].

##### Mechanism

It is possible that feeling listened to and being able to share likely enhances mood related outcomes as a direct result of the session [41,42]. This may allow or enhance the individual’s ability to reflect on their own story and experience.

##### Reported Outcome: Fear

Engaging in the intervention seemed to be beneficial in reducing fear after the stroke and instilling a feeling of reassurance [31,32]. It should be noted however that positive changes were noted by Corsten et al. [41,42] but they were not significant (*p* = 0.30; *p* = 0.50, respectively). Corsten et al. [41] identified a moderate effect size (*d* = 0.56), but Corsten et al. [42] identified no effect size (*d* = 0.06). Appalasamy et al. [40] reported that the narrative intervention helped reduce fear and enable participants to overcome the challenges created by stroke.

##### Mechanism

Clark et al. [29] explored stroke survivors’ perceptions of the potential challenges of a self-management group prior to attendance. Participants felt that certain aspects of the groups, such as sharing experiences of stroke and recalling distressing memories, might be difficult and “distressing to hear” for some individuals. The potential emotional impact of the intervention on the peer supporters was identified by programme organisers, who thought that it might mean reliving the distress related to earlier experiences of stroke, thus increasing fear, worry or anxiety [32]. Appalasamy et al. [40] suggested that confidence was raised through the intervention which likely countered the experience of fear.

#### 3.4.5. Theme 5: Enhanced Awareness of Stroke and Knowledge

##### Reported Outcome

Seeking and receiving knowledge was one reason that stroke survivors gave for attending the groups [30,31]. Appalasamy et al. [36] found significantly increased knowledge of stroke (F = 11.54, *p* < 0.001). Stroke survivors and their partners of care received valuable informational support from the interventions, about stroke prevention and recovery, finding and accessing stroke support services in the community [30,31,32,34]. This type of information, related to living with stroke in the community and provided by peers, was considered more helpful than that offered by HCPs during hospitalisation [31,32]. One participant considered the information provided by a stroke survivor who had “*substantial experience subsequent to their stroke*” as particularly valuable [34]. Peers provided experiential knowledge and information derived from real lived experience, which facilitated relatedness [32,34]. One study [39] quantified this by reporting that a majority of participants (n = 10/13) agreed or strongly agreed with the statement “*I know more about community and stroke resources*” after attending the peer support group, and that the same number agreed or strongly agreed that the group widened their knowledge of condition, post-stroke recovery and opportunities such as voluntary jobs and modified leisure activities. Participants in the Christensen et al. [30] study asserted that information related to stroke and recovery from stroke was difficult to find and that they and their carers significantly valued the opportunity to learn from both peers and experts.

##### Mechanism

Social comparison and listening to stories as well as more direct advice using social control may well have existed within this theme and acted as a mechanism [40].

## 4. Discussion

The current review has demonstrated that narrative or peer sharing intervention can positively impact important psychosocial outcomes and this is most often due to social comparison and social control. This integrated review was based on a limited number of studies with small sample sizes, which might lead to an underestimation of the findings reported. The current recommendations focus on the findings with the highest strength of evidence which suggested that the intervention most often can; introduce a sense of positivity and hope for individuals, reduce loneliness, and enable coping through knowledge exchange. The reduction in loneliness is supported by past evidence [8]. In addition, past reviews have identified that storytelling interventions can enable coping and enhance knowledge [10] and provide an opportunity to express and understand true experiences and feelings related to the illness [10,23]. Additionally, storytelling has been reported to reduce psychological distress and allowed fears and worries to be discussed [23].

Social and internal mechanisms identified within the current review have also been identified and supported by past review evidence in chronic illnesses. For instance, self-reflection has been identified as a key process and mechanism through which positive impacts of storytelling can be found, as it allows participants to create and re-create plots of their own story [8,10,23]. Further to this, social comparisons can help change the perception of their own situation and make the individual feel less isolated [8]. It may be that relatedness with others enables a therapeutic environment generated by enhanced trust [8,23], where participants feel that their experiences are legitimized [23].

It was useful to evaluate the interventions in both one-to-one and group formats, as the two formats created different experiences. The one-to-one format could be tailored to the individual needs and difficulties occurring in the post-stroke trajectory; however, it would be less likely to be cost-effective. The group format could be less expensive and there are mutual advantages when members exchange their stories related to illness and health, potentially leading to the learning and examination of new information, stroke-related resources, practical coping skills and strategies [43]. The National Institute for Health and Clinical Excellence [44] identifies group participation as potentially beneficial for people with chronic conditions including stroke. Therefore, it might be an appropriate format for providing a storytelling intervention, in comparison with the one-to-one format. Within the current review it is possible that smaller group sizes may be more optimal to promote the benefits of sharing and feeling heard. Greater understanding of the impact of social comparison is needed and how it may influence outcomes. For instance, downward comparison (comparing oneself to those perceived as being worse off) can have a positive impact on the individual appraisal of their own well-being [45,46], whereas upward social comparison (comparing oneself to those perceived as better off who face a similar problem) has been associated with less life satisfaction when compared with downward comparisons [47]. Research [46] has also highlighted that people making downward social comparisons can be less likely to seek support for their own needs. Further research is required to evaluate if upward social comparisons are useful and how to introduce and optimize the benefits from downward social comparisons. Further research also needs to be able to consider if, how and when social control is used for peers.

Future research is required to consider if a storytelling group should include both stroke survivors and their carers or families. The three qualitative studies exploring this aspect of interventions [29,33,34] consistently identified the benefits of involving carers or families in the group sessions such as informational exchange and support. This is supported by past research [48], which identified such support as essential. Further research is required to consider the impact of the intervention on mood states, as changes have been identified across other chronic illnesses [23]. Additionally, consideration is needed around the impact of how close the diagnosis was or how severe the symptoms were as these aspects may be important in identifying if and how changes occur [49].

### 4.1. Limitations of the Studies

The heterogeneity of the interventions regarding type, design and personnel may have had an adverse effect on their implementation in clinical practice. There was a large range of time since stroke for participants across studies and further research must consider time since stroke as an aspect to explore further. Methodological limitations identified as having contributed to a decrease in overall quality included lack of randomization and blinding of personal assessors. Only two studies were a randomized control trial and further well-designed studies are needed. Studies did not consider the impact of physiological parameters of participants which are important as they may impact on the experience of the intervention, for instance higher blood pressure has been associated with greater cognitive decline [50]. It was difficult to consider the exact instances of social control as against social comparison from the literature. Limitations apparent in the qualitative studies were related to study design, lack of reflexivity, data analysis and the reporting of findings. The findings may be transferable to other stroke settings, whilst not being generalizable.

### 4.2. Implications

Storytelling and peer sharing appear to be an useful intervention to enhance positive emotions and offer hope in a way that empowers people with stroke. It should be considered as an accessible method to encourage positive psychosocial adaptation. Peer sharing and storytelling should be considered in different formats including one-to-one, group settings and e-formats; for instance, the UK The Stroke Association’s website provides stroke stories (stroke association https://www.stroke.org.uk/stroke-stories, accessed on 12 May 2021). It is important that the participant preference be accounted for by giving the individual choice about attendance and also identify if, when and how storytelling is used. Clinicians and researchers need to be mindful of instances when the group setting could be less useful (i.e., when a social comparison could be considered out of reach and impossible to achieve). Further research needs to consider the value and use of sharing the group with their family members or carers. Finally, future research should consider how hope and emotions can be captured to identify change created by storytelling and further well-powered and well-designed trials are needed.

## 5. Conclusions

To conclude, the current evidence appears to be positive and identify positive psychosocial outcomes gained from peer group interventions. It is important to recognize that peers provide an essential role for knowledge exchange and an ability to reduce loneliness and should be (within the limits of the implications section) considered as part of rehabilitation following a stroke. Future research is required to confirm the effectiveness and therapeutic value of peer-led storytelling interventions in promoting the psychosocial well-being of stroke survivors.

## Figures and Tables

**Figure 1 behavsci-11-00088-f001:**
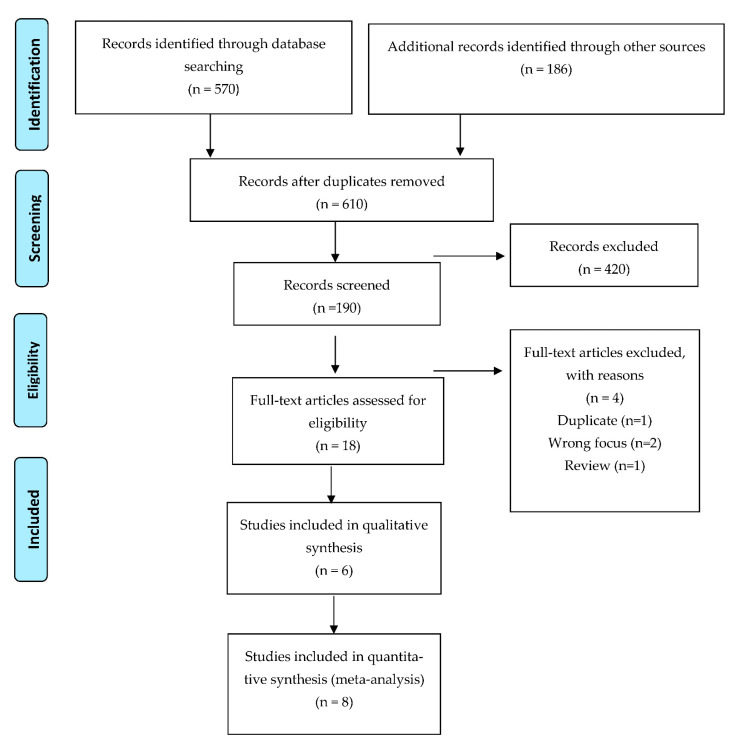
PRISMA 2009 Flow Diagram.

**Table 1 behavsci-11-00088-t001:** Summary of evidence.

Theme	Level of Evidence [21] for Quantitative Evidence and CerQual Confidence [22] of Evidence	Summary of Evidence	Suggested Mechanisms
Introducing the concept of hope and learning to be positive	1 level II study 5 qualitative studies with high CerQual confidence of evidence	It is very likely that the intervention can introduce hope	Social comparison Social control
The enhanced ability to cope	3 Level IV studies 1 qualitative study with low CerQual confidence of evidence	It is likely the intervention can enhance the individual ability to cope	Feeling understood and heard by others Social comparison Social control Self-efficacy
The impact of loneliness and social interaction	1 Level IV studies 8 qualitative studies with high CerQual confidence in evidence	It is likely that peer support and the development of friendship can impact isolation and loneliness	Feeling understood and heard by others Validated sense of suffering Social comparison Social control
Impact on emotions; depression and related emotions	1 level II study 2 Level IV studies 2 qualitative studies with low CerQual confidence of evidence	It is not yet possible to determine the impact of the intervention on mood states. Well powered studies are needed to consider if the statistically insignificant but very large effect sizes identified on negative mood states (confusion, anger, sadness and tiredness) can be repeated.	Feeling understood and heard by others Validated sense of suffering
Impact on emotions; fear	2 Level IV studies 1 qualitative study with low CerQual confidence of evidence	It is unlikely that the intervention impacted fear	Feeling understood and heard by others
Enhanced awareness of stroke	1 level II study 1 Level IV studies with low CerQual confidence of evidence	It is likely that the intervention enhanced understanding of stroke	Social comparison Social control

## Data Availability

Data available in the Appendix A.

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
