# Peer review of "An Integrative Review Considering the Impact of Storytelling and Sharing Interventions in Stroke"

_behavsci, 2021, doi:10.3390/bs11060088_

Round 1

Reviewer 1 Report

  1. This is a sound methodological integrative review of this topic.

  1. Can author provide an explanation to why, “The interventions had to be recorded on or evaluated through any methods, such as digital (online-based) and verbal (one-to-one, in groups or telephone-based.” As noted later in the methods section again: “it had to record the effect or perceived value and experience of engaging in the intervention…”

Does that mean only interventions that were recorded were included? It is a bit unclear? And if so why, did the intervention need to be recorded.

3.      “Studies in other languages were translated to English. Studies in other languages were excluded for lack of a translation service.” These statements are unclear. Perhaps just include the latter phrase and exclude the first.

  1. When this comment is made in the discussion section, “The current recommendations focus on the findings with the highest strength of evidence which suggested that the intervention most often can…..”. Can the author elaborate on how they focused more on the findings of studies with the highest strength? Especially since if there was corroborating evidence across studies. I think this would be good to elaborate upon in the methods section in how they decided to focus on the high-quality studies.

  1. Did the authors find any differences in themes between the one-on-one format versus group format? Nothing was mentioned in the results section but than later discussed in the discussion section.

  1. Can the author elaborate on the concept of social control in the discussion and why it related to the review themes reported? As there was to some extent about social comparison. The discussion needs a bit more work so that is positions the authors findings with the current literature.

Author Response

Reviewer 1

Thank you for your comments and suggestions.

  1. This is a sound methodological integrative review of this topic.

 AS: Thank you for your comment.

  1. Can author provide an explanation to why, “The interventions had to be recorded on or evaluated through any methods, such as digital (online-based) and verbal (one-to-one, in groups or telephone-based.” As noted later in the methods section again: “it had to record the effect or perceived value and experience of engaging in the intervention…”

 AS: this has been updated and clarified.

Does that mean only interventions that were recorded were included? It is a bit unclear? And if so why, did the intervention need to be recorded.

AS: This was a typographical error. We have clarified this point now. Thank you.

  1. “Studies in other languages were translated to English. Studies in other languages were excluded for lack of a translation service.” These statements are unclear. Perhaps just include the latter phrase and exclude the first.

AS: this has been updated and should not have had the latter point.

  1. When this comment is made in the discussion section, “The current recommendations focus on the findings with the highest strength of evidence which suggested that the intervention most often can…..”. Can the author elaborate on how they focused more on the findings of studies with the highest strength? Especially since if there was corroborating evidence across studies. I think this would be good to elaborate upon in the methods section in how they decided to focus on the high-quality studies.

 AS: thank you for this comment. We have separated out the criteria for assessing strength and added in a qualitative assessment of strength of evidence too.

  1. Did the authors find any differences in themes between the one-on-one format versus group format? Nothing was mentioned in the results section but than later discussed in the discussion section.

 AS: Line 315 identifies this information in a supplementary file. We have expanded this to make sure the reader is aware of these results. To paste them into this section could create a very large results section so we opted for the supplementary file.

  1. Can the author elaborate on the concept of social control in the discussion and why it related to the review themes reported? As there was to some extent about social comparison. The discussion needs a bit more work so that is positions the authors findings with the current literature.

AS: we have elaborated on social control and provided additional considerations.

Reviewer 2 Report

Summary: The current manuscript systematically review the impact of storytelling and sharing interventions in stroke.

The authors show that storytelling interventions appear to be associate to a positive environment which can enable hope and coping mechanisms. Although authors present interesting findings, some aspects could be improved.

Introduction: Overall, the introduction provides a broad background and rationale for the research. However, other important aspects associated with cardiovascular disorders and especially stroke (such as effects on cognitive function) are not mentioned. I suggest including them as they may provide additional motivation and give a different impact of this systematic review. For Example I suggest:

Forte, G., & Casagrande, M. (2020). Effects of Blood Pressure on Cognitive Performance in Aging: A Systematic Review. Brain Sciences10(12), 919.

Methods: The method is comprehensive. I suggest to better define Inclusion and Exclusion criteria (maybe a list could be help the reader). I suggest to restructure the research paragraph (2.3), the steps confuse the reader. Also, who was in charge of the research? were the concerns discussed? who made the decisions? Did anyone supervise the research?

Analysis: the analyses are well conducted.

Results: The summary of the study provided is well-defined and fits according to the analysis plan provided. However this paragraph is confusing in some parts, the choice to report the results in this way could be structured to make clear what emerges from the literature. Moreover, I suggest to restructure Table 1, correct some typos (e.g., line 251) and make this table clearer (especially in “Summary of Evidence”). I suggest to better discuss risk of bias (maybe a single paragraph called “risk of Bias” could be helpful).

Discussion: the discussion appear to be a summary of the results, I suggest reporting the usefulness of this study and further perspective.

General comment: I would also encourage the authors to check all references and to proofread the manuscript to improve the English language.

Author Response

Reviewer 2

Introduction: Overall, the introduction provides a broad background and rationale for the research. However, other important aspects associated with cardiovascular disorders and especially stroke (such as effects on cognitive function) are not mentioned. I suggest including them as they may provide additional motivation and give a different impact of this systematic review. For Example I suggest:

Forte, G., & Casagrande, M. (2020). Effects of Blood Pressure on Cognitive Performance in Aging: A Systematic Review. Brain Sciences, 9, 34; http:www.doi.org/10.3390/jcm9010034

, 919.

 AS: we acknowledged this point but placed it within the limitations section. 

Methods: The method is comprehensive. I suggest to better define Inclusion and Exclusion criteria (maybe a list could be help the reader).

AS: eligibility criteria is nearly always presented as PICOS or alternative for reviews so we have not changed this aspect as it would deviate from a recommended standard.

I suggest to restructure the research paragraph (2.3), the steps confuse the reader.

AS: We agree and have removed the steps.

 Also, who was in charge of the research? were the concerns discussed? who made the decisions? Did anyone supervise the research?

AS: we have added the roles of people within the methods section which provides this detail. 

Analysis: the analyses are well conducted.

Results: The summary of the study provided is well-defined and fits according to the analysis plan provided. However this paragraph is confusing in some parts, the choice to report the results in this way could be structured to make clear what emerges from the literature. Moreover, I suggest to restructure Table 1, correct some typos (e.g., line 251) and make this table clearer (especially in “Summary of Evidence”). I suggest to better discuss risk of bias (maybe a single paragraph called “risk of Bias” could be helpful).

AS: thank you for these suggestions we have updated this section.

Discussion: the discussion appear to be a summary of the results, I suggest reporting the usefulness of this study and further perspective.

 AS: thank you we have made changes to consider further discussion and consideration of the implications.

General comment: I would also encourage the authors to check all references and to proofread the manuscript to improve the English language.

AS: thank you this has been undertaken.

Reviewer 3 Report

There are some minor grammatical issues and spacing issues. 

Please be more descriptive in the abstract. I have offered some suggestions on the manuscript.

In the introduction, distinguish storytelling from peer sharing.

There is a lack of clarity around the inclusion of studies in other languages.

Could you provide a rationale for including participants from an acute stroke phase to 174 months, and for the time period (2009-2019/2020)?

I am unsure how marital status and residential arrangement is relevant to the study findings. 

In the results section, you seem to include some of your (authors) own assumption. Ensure that the results are grounded in the data. 

Line 358 talks about participants revisiting and revising their own story. Can you elaborate on this?

While the review demonstrate narrative or peer sharing intervention can  positively impact important psychosocial outcomes, can you identify potential negative impacts of group-based story telling (e.g.,  social comparison)?

Elaborate on the conclusion. Summarize the findings and the larger implications of the study, and demonstrate the importance of your ideas. 

Author Response

Reviewer 3

There are some minor grammatical issues and spacing issues. 

AS: thank you we have attended to these issues.

Please be more descriptive in the abstract. I have offered some suggestions on the manuscript.

AS: we have increased this section to provide details of all the sections.

In the introduction, distinguish storytelling from peer sharing.

AS: this has been done.

There is a lack of clarity around the inclusion of studies in other languages.

AS: thank you this has been updated. There was a typographical error.

Could you provide a rationale for including participants from an acute stroke phase to 174 months, and for the time period (2009-2019/2020)?

AS: no. We check the searches for articles without a lower limit.

I am unsure how marital status and residential arrangement is relevant to the study findings. 

AS: social support and living arrangements will impact aspects like perceptions of isolation.

In the results section, you seem to include some of your (authors) own assumption. Ensure that the results are grounded in the data. 

AS: Thank you for this comment we have removed one statement and validated two others with references to make sure data is based in literature.

Line 358 talks about participants revisiting and revising their own story. Can you elaborate on this?

AS: we have elaborated on this.

While the review demonstrate narrative or peer sharing intervention can  positively impact important psychosocial outcomes, can you identify potential negative impacts of group-based story telling (e.g.,  social comparison)?

AS: we have expanded this.

Elaborate on the conclusion. Summarize the findings and the larger implications of the study, and demonstrate the importance of your ideas. 

AS: we have undertaken this.

Round 2

Reviewer 2 Report

the manuscript is substantially improved